# Detailed Modeling of Surface-Plasmon Resonance Spectrometer Response for Accurate Correction

**DOI:** 10.3390/s25030894

**Published:** 2025-02-01

**Authors:** Ricardo David Araguillin-López, Angel Dickerson Méndez-Cevallos, César Costa-Vera

**Affiliations:** 1Departamento de Automatización y Control Industrial, Escuela Politécnica Nacional, Quito 170525, Ecuador; 2Mass Spectrometry and Optical Spectroscopy Group, Departamento Física, Escuela Politécnica Nacional, Quito 170525, Ecuador; angel.mendez@epn.edu.ec (A.D.M.-C.); cesar.costa@epn.edu.ec (C.C.-V.)

**Keywords:** modeling, transfer function, surface-plasmon resonance, Kretschmann configuration, spectroscopy

## Abstract

This work identifies and models the inline devices in an experimental surface-plasmon resonance spectroscopy setup to determine the system’s transfer function. This allows for the comparison of theoretical and experimental responses and the analysis of the dynamics of the components of an analyte placed on the sensor at the nanometer scale. The transfer functions of individual components, including the light source, polarizers, spectrometer, optical fibers, and the SPR sensor, were determined experimentally and theoretically. The theoretical model employed Planck’s law for the light source, manufacturer specifications for some components, and experimental characterization for others, such as the polarizers and optical fibers. The SPR sensor was modeled using characteristic matrix theory, incorporating the optical constants of the prism, gold film, chromium adhesive layer, and analyte. The combined transfer functions created a comprehensive model of the entire experimental system. This model successfully reproduced the experimental SPR spectrum with a similarity greater than 95%. The system’s operational range was also extended, constrained by the signal-to-noise ratio at the spectrum’s edges. The detailed model allows for the accurate correction of the measured spectra, which will be essential for the further analysis of nanosuspensions and molecules dissolved in liquids.

## 1. Introduction

Surface-plasmon resonance (SPR) is a well-established technique for the real-time detection and analysis of biomolecular interactions. Its applications span diverse fields, including molecular interaction analysis, biomarker detection, molecular imaging, electrochemical studies, and food analysis [1,2]. Recent advances have extended SPR to more complex systems, such as nanoparticles and nanomaterials [3,4]. However, accurately modeling real SPR systems remains challenging [5]. Existing models, often based on simplified assumptions and focusing primarily on the SPR sensor itself, frequently fail to adequately account for the influence of instrumental factors [6,7]. This limitation restricts their applicability to complex experimental setups. Addressing these limitations requires a comprehensive approach that considers both the underlying physical principles of SPR and the specific experimental setup configuration.

Surface plasmons (SPs) are collective oscillation modes induced by evanescent electromagnetic waves on the free-electron charge density at the interface between a metal and a dielectric medium. Plasmonic oscillations are typically initiated by the incidence of a beam of light on the surface of the metal [8]. Surface-plasmon resonance spectroscopy (SPR-S) leverages this phenomenon to quantify molecular interactions occurring in a medium in contact with the metal surface [9]. A typical experimental setup, based on the Kretschmann configuration [10], employs a high-refractive-index prism coated with a thin gold film, along with a light source, a polarizer, and a spectrometer (Figure 1). Incident *p*-polarized light at a specific angle (θ) excites surface plasmons. Resonance occurs when the momentum of the incident light, $k_{x}$, equals the momentum of the surface plasmons $k_{SP}$ [11] described by:(1)$$n_{p}\sin\left(\theta \right)=\sqrt{\frac{\varepsilon_{m}\varepsilon_{a}}{\varepsilon_{m}+\varepsilon_{a}}},$$
where $n_{p}$, $\varepsilon_{m}$, and $\varepsilon_{a}$ are the refractive index of the prism and the complex dielectric constants for the metal and the analyte, respectively [8].

Upon resonance, a unique spectral signature is created for the analyte [12]. The resonance wavelength provides information about molecular interactions at the nanometer scale [5]. However, the measured spectrum is susceptible to instrumental influences, including the wavelength-dependent radiance of the light source, attenuation in optical fibers, the transmittance of the polarizer, and other optical components (e.g., lenses, collimators), as well as the detection efficiency of the spectrometer [5,7,13,14]. These factors can shift the observed resonance wavelength, making accurate determination challenging [5,13,14]. Accurate interpretation requires the correction of the measured spectrum for these instrumental effects [14]. Current correction methods include polynomial or Lorentzian fitting [12], identifying the intensity minimum [15,16], calculating the spectral centroid [11], and compensating for the detector response of the spectrometer [7]. Liu et al. present a spectral correction method for surface-plasmon resonance (SPR) measurements that compensates for the wavelength-dependent response efficiency of the spectrometer’s detector. Recognizing that detector response can introduce spectral distortions affecting accurate resonance wavelength extraction, they propose distinct correction models optimized for silicon (Si) and indium gallium arsenide (InGaAs) detectors, comparing them to a routine normalization approach. Their methods aim to improve spectral symmetry (SMT), resonance depth (DRD), and full width at half maximum (FWHM), enabling a more accurate determination of the resonance wavelength directly from the spectrum, without complex fitting procedures. These metrics are chosen as they provide a quantitative assessment of spectral shape and features, directly impacted by detector response variations [7].

Although the concept of transfer function (TF) is widely utilized in electronics and control systems for modeling input–output relationships, its application to the detailed modeling and spectral correction of SPR spectrometers, to the best of our knowledge, has not been previously explored in the literature. A TF quantifies how a system modifies incident light as a function of wavelength, represented in the frequency domain as H=Y/X, where *X* is the input and *Y* is the output [17]. This work introduces a novel approach that employs TFs to characterize and correct the spectral response of an SPR spectrometer, taking into account the individual contributions of each optical component. The total transfer function of the system is the product of the individual component TFs.(2)$$H_{TOTAL}\left(\lambda \right)=H_{1}\left(\lambda \right)\;H_{2}\left(\lambda \right)\;...\;H_{n}\left(\lambda \right),$$
where $H_{TOTAL}\left(\lambda \right)$ is the total transfer function, $H_{i}\left(\lambda \right)$ is the transfer function of the *i*-th component, and λ is the wavelength. It is proposed that this TF-based model will accurately reproduce the experimental spectrum, thereby enabling the precise correction of measured SPR spectra and facilitating an improved analysis of molecular interactions.

This work presents a detailed model of a homemade SPR-S system, incorporating the transfer functions of individual components to reproduce the system’s spectral response accurately. Section 2 describes the experimental setup, and the process of determining the TF for individual components. Section 3 validates the model by comparing theoretical and experimental responses. Section 4 describes the possible noise sources affecting the TFs and mitigation strategies. Section 5 demonstrates the model’s application for correcting SPR spectra. Finally, Section 6 presents the conclusions and outlines future research directions.

## 2. SPR Spectrometer System: Components and Transfer Function Determination

The SPR spectroscopy system used in this work includes a light source, a polarizer, an SPR sensor, optical fibers, and a spectrometer, as shown in Figure 2. The transfer function of each component can be determined by modeling the components separately. This section describes the theoretical framework used in the SPR spectrometer system and details the method used to determine the transfer function of each element. The models evaluated for each device are detailed in the following subsections, and the most applicable models are selected based on their correlation with the experimental results. The individual transfer functions will then be combined to create a comprehensive model of the entire SPR system.

### 2.1. Spectrometer

The spectrometer plays a critical role in shaping the recorded spectrum because the light collected by the SPR sensor passes through its internal components, each of which has a different spectral response. Our system uses a compact spectrometer (CCS200, Thorlabs Inc., Newton, NJ, USA) consisting of a 600 lines/mm diffraction grating and a CCD linear sensor (TCD1304DG, Toshiba, Minato, Japan). Each component contributes to the spectrometer’s overall transfer function ($H_{Spec}\left(\lambda \right)$) affecting the observed signal.

*Diffraction Grating*: This device disperses incoming light according to its wavelength. The absolute efficiency, G(λ), provided by the manufacturer [18], represents its capability to diffract light toward the CCD sensor as a function of wavelength. This efficiency curve acts as the transfer function of the grating.*CCD Sensor*: The dispersed light falls onto the CCD linear sensor, which converts the incident photons into an electrical signal. Its relative responsivity curve, S(λ) [19], describes the sensor’s wavelength-dependent sensitivity to light and effectively acts as its transfer function.

The overall transfer function of the spectrometer, $H_{Spec}\left(\lambda \right)$, is the product of the individual transfer functions of the diffraction grating and the CCD sensor: $H_{Spec}\left(\lambda \right)=G\left(\lambda \right)S\left(\lambda \right)$ (Figure 3). This product is appropriate because the light first interacts with the grating and then with the CCD; they are sequential elements in the optical path. The careful characterization of this transfer function is essential as it provides the basis for determining the transfer functions of all other components and for correcting the acquired SPR spectra.

### 2.2. Light Source

In the system, a stabilized tungsten–halogen lamp (SLS201L, Thorlabs Inc., Newton, NJ, USA) is used to illuminate the gold film. The lamp’s emission spectrum, provided by the manufacturer (Figure 4, red line) [20], is expected to follow Planck’s blackbody radiation law, which describes energy emission as a function of temperature and wavelength:(3)$$I\left(\lambda ,\;T\right)\;=\;\frac{2\pi \;h\;c^{2}}{\lambda^{5}}\frac{1}{e^{\frac{hc}{\lambda \;k_{B}T}}-1}.$$
However, the recorded spectrum (Figure 4, green line) deviates from the expected profile due to the transfer function of the spectrometer ($H_{Spec}$), which could affect the accuracy of the data. Fitting Planck’s law to the lamp spectrum yields an optimum blackbody temperature of 2650 K for wavelengths between 300 and 1000 nm. The fitting was performed using curve_fit from the scipy.optimize module in Python. The fit achieved a coefficient of determination $R^{2}$ of 0.9995. This fitted curve provides a theoretical model of the lamp’s performance X(λ) and enables accurate spectrometer calibration.

### 2.3. Polarizer

To achieve plasmon resonance, *p*-polarized light is required. Our experimental setup includes a polarizer [LPVISE050-A, Thorlabs Inc., NJ] to adjust the polarization of the incident light. While the manufacturer specifies its performance for wavelengths between 400 to 700 nm [21] its influence extends beyond this range. To broaden the spectral analysis, the transfer function of the polarizer, P(λ), was experimentally characterized by using a Lightsource DH-mini lamp [UV-VIS-NIR, Ocean Optics, Orlando, FL, USA]. By measuring both the incident and transmitted light intensities along the polarization axis, and accounting for the TF of the spectrometer, $H_{Spec}$, the polarizer’s transmittance was determined over the 350–1000 nm range. A Savitzky–Golay filter (window size = 15, polynomial order = 3) was applied to smooth the resulting P(λ) curve (see Figure 5).

### 2.4. SPR Sensor

The SPR sensor, based on the Kretschmann configuration, consists of an SF11 glass prism coated with a 50 nm gold film and a 0.2 nm chromium adhesive layer (HORIBA France SAS, Lyon, France). The reflectivity of this bimetallic structure was modeled using characteristic matrix theory [22]. This theory allows us to calculate the propagation of the electromagnetic light wave through an *N*-layer system, where each layer interacts directly with any other layer or optical element. Consequently, the system is modeled by a matrix product *M* linking every layer, represented by the matrix $U_{k}$, sequentially in a non-commutative way, as follows(4)$$M=\prod_{k=2}^{N-1}U_{k}=\prod_{k=2}^{N-1}\;\left(\begin{array}{cc} \cos\beta_{k} & \left(-i\sin\beta_{k}\right)/q_{k}\\ -iq_{k}\sin\beta_{k} & \cos\beta_{k}. \end{array}\right)$$
where(5)$$\beta_{k}=\frac{2\pi d_{k}}{\lambda }{\left(\varepsilon_{k}-n_{1}^{2}\sin^{2}\theta \right)}^{\frac{1}{2}},$$(6)$$q_{k}=\frac{{\left(\varepsilon_{k}-n_{1}^{2}\sin^{2}\theta \right)}^{\frac{1}{2}}}{\varepsilon_{k}},$$
According to Fresnel’s equations, the reflectivity $R_{p}$ and reflection coefficient $r_{p}$ for p—polarized light are given by(7)$$R_{p}={\left|r_{p}\right|}^{2}={\left|\frac{\left(M_{11}+M_{12}q_{N}\right)q_{1}-\left(M_{21}+M_{22}q_{N}\right)}{\left(M_{11}+M_{12}q_{N}\right)q_{1}+\left(M_{21}+M_{22}q_{N}\right)}\right|}^{2}.$$
In these equations, subindex 1 refers to the high refractive index prism, while indices 2 through N−1 indicate the intermediate layers situated between the prism and the analyte layer, designated as *N*. In this context, the term $\varepsilon_{k}$ represents the complex dielectric function of the *k*-th layer, while $beta_{k}$ denotes a parameter associated with its optical path, with $d_{k}$ denoting the layer thickness. The variable $q_{k}$ acts as a form parameter, with $n_{1}$ corresponding to the refractive index of the prism and θ denoting the angle of incidence of light on the interface between the prism and the first intermediate layer.

To evaluate Equation (Equation 7), it is necessary to input the optical constants of the materials that comprise the SPR sensor. Because metals absorb light, their permittivity is expressed as a complex number. The complex refractive index is defined as the square root of the dielectric constant, where the real part *n* is the refractive index and the imaginary part κ is the absorption coefficient [22]. The experimental refractive indices required for the prism [23], gold [24], chromium [25], and water [26] are shown in Figure 6. Although the refractive indices of the prism and water appear to be constant, they are, in fact, subject to slight variation within the specified spectral range. The aforementioned curves facilitate the attainment of a more precise spectral response for the sensor.

### 2.5. Optical Fibers

The system employs two high-quality multimode optical fibers to guide the light efficiently. The first optical fiber [P400-2-UV/VIS, Ocean Optics BV] carries the light from the source to the polarizer and then strikes the sensor. The reflected light by the sensor is then guided by the second optical fiber [FG050LGA, Thorlabs Inc., NJ] to the spectrometer.

To determine the transfer functions, it was essential to examine their attenuation spectra α [27,28]. From this, the transmittance of both optical fibers was calculated using the ratio (Equation 8), which is based on the reduction in light intensity as it propagates through the fiber over a length *L*. The length of the optical fibers is one meter. This approach was employed to model the transfer functions $FO_{1}\left(\lambda \right)$ and $FO_{2}\left(\lambda \right)$. Figure 7 illustrates, on the left, the attenuation spectra of the fibers and, on the right, their transfer functions.(8)$$Transmittance=\;10^{-\alpha L/10}.$$

### 2.6. Other Optical Elements

The remaining optical components used in this system, such as the prism, lens, and collimators, located along the light path, are made from highly transparent optical glasses. It is assumed that the transmittance, or transfer function, of these materials is unity within the optical spectral range, due to the negligible absorption and scattering of the materials in this region. Accordingly, a detailed characterization of these components was deemed unnecessary.

The overall transfer function of the SPR system, $H_{Total}\left(\lambda \right)$, is then calculated as the product of the individual transfer functions described above, as given by Equation (Equation 2) in the Introduction. This combined transfer function will be used in the next section to model and validate the complete system response against experimental measurements.

## 3. Experimental Modeling Validation

To build an accurate system model, it is essential to verify its transfer function (TF) through experimental data. This section describes the validation process, focusing on the interaction of the light source, polarizer, optical fibers, and spectrometer without the SPR sensor. This simplified approach allows for a clearer assessment of the model’s ability to reproduce the experimental spectral response and isolate the effects of these components before incorporating the more complex SPR sensor interaction. Three models of increasing complexity were developed and compared to the experimental results.

*Model 1:* This model served as a baseline and assumed a negligible influence of the optical fibers employed in the setup due to their relatively short length. The overall transfer function of this model is:(9)$$TF_{M1}\left(\lambda \right)=X\left(\lambda \right)\;P\left(\lambda \right)\;H_{Spec}\left(\lambda \right),$$
where X(λ) is the spectral output of the light source, P(λ) is the transmittance of the polarizer, and $H_{Spec}\left(\lambda \right)$ is the spectrometer’s transfer function, encompassing the effects of both the grating and the CCD sensor. This model represents a simplified scenario where the fiber’s contribution is considered minimal.*Model 2:* This model incorporates the transfer functions associated with the optical fibers, in order to quantify the effect they have on the spectral response of the system. The complete transfer function is mathematically represented as follows:(10)$$TF_{M2}\left(\lambda \right)=X\left(\lambda \right)\;FO_{1}\left(\lambda \right)\;P\left(\lambda \right)\;FO_{2}\left(\lambda \right)\;H_{Spec}\left(\lambda \right).$$In this equation, $FO_{1}\left(\lambda \right)$ and $FO_{2}\left(\lambda \right)$ represent the transfer functions of the illumination and collection fibers, respectively. By integrating these elements, Model 2 accounts for the wavelength-dependent attenuation characteristic of the fibers and, at the same time, allows for a more thorough examination of the system behavior under varying conditions.*Model 3:* The enhanced model is based on the core concepts outlined in Model 2 and includes a convolution operation to account for the combined effects of chromatic aberrations, spatial disturbances, and wavelength-dependent variations in the refractive index within the optical components. As described by Pérez Tudela [29], convolution is a mathematical tool that describes the modification of a signal as it passes through a system. In our case, each optical component modifies the spectrum of the incident light. These modifications can be represented as convolutions with the transfer function of each component.Chromatic aberrations, arising from the wavelength-dependent refraction of light by lenses and other optical components, cause a spatial dispersion of the spectral components. This dispersion is modeled as a convolution with a function representing the point spread function of the system for each wavelength [30].Furthermore, spatial disturbances, such as beam divergence, scattering, and imperfections in optical surfaces, contribute to the spatial diffusion of light. This diffusion is also modeled as a convolution, where the scattering function represents the spatial distribution of light after interacting with the optical component [31].Finally, the dependence of the refractive index on the wavelength within the optical components alters the speed of light propagation, modifying the spectral distribution. This modification is represented as a convolution, with a function describing the spectral phase change induced by the change in refractive index [32].Model 3 combines these contributions through successive convolutions with the transfer functions of each component, as expressed in Equation (Equation 11). The convolution function used is ‘np.convolve’ from the ‘NumPy’ library in Python. The mathematical representation of Model 3 is articulated as follows:(11)$$TF_{M3}\left(\lambda \right)=X\left(\lambda \right)*FO_{1}\left(\lambda \right)*P\left(\lambda \right)*FO_{2}\left(\lambda \right)*G\left(\lambda \right)*S\left(\lambda \right)$$The convolution operation provides a more realistic representation of the complex interactions between the optical elements, considering the spectral broadening and blurring effects that can occur. It accounts for how the spectral output of each component is modified and spread out by the subsequent components in the optical path. This comprehensive approach enhances the model’s ability to handle complex optical phenomena and ensures a higher accuracy in the representation of light interactions within the system.

The experimental setup for model validation consisted of a stabilized tungsten–halogen lamp (SLS201L, Thorlabs Inc., NJ) as the light source, X(λ), operating at a color temperature of 2650 K. Light from the source was coupled into a 1-m P400-2-UV/VIS optical fiber (Ocean Optics BV), passed through the polarizer (LPVISE050-A, Thorlabs Inc., NJ), and then coupled into a 1-meter FG050LGA collection fiber (Thorlabs Inc., NJ) leading to the spectrometer (CCS200, Thorlabs Inc., NJ). The spectrometer’s integration time was set to 8 ms. The experimentally measured spectrum, $Y_{E}\left(\lambda \right)$, was then compared to the spectra predicted by each model. To quantify the degree of agreement between the experimental and theoretical spectra, the mean square error (MSE) and the coefficient of determination ($R^{2}$) were employed as figures of merit.

Figure 8 shows the normalized experimental spectrum and the spectra generated by the three models. Model 1, which excludes the influence of fibers, accounts for an $R^{2}$ of 0.8689 and an MSE of 0.0150, with discrepancies primarily at the spectral extremes. It serves to illustrate the limitations of this simplified model. Model 2, which incorporates fiber influence, significantly improved the agreement with the experimental data, achieving an $R^{2}$ of 0.9630 and an MSE of 0.0041. This improvement serves to confirm the importance of including the fiber transfer functions. Model 3, which incorporates a convolution operation, further refined the agreement, resulting in an $R^{2}$ of 0.9842 and an MSE of 0.0018. This validated Model 3 will be used in the subsequent section for correcting measured SPR spectra. The remaining discrepancies are likely within the experimental uncertainty, based on the combined effects of light source stability, low signal-to-noise ratios in spectrum regions, temperature fluctuations, and factors not explicitly modeled, such as dust or small imperfections in the optical components. A detailed account of the noise sources considered and the mitigation strategies employed is provided in Section 4, which resulted in low uncertainties.

## 4. Noise Sources and Mitigation Strategies

To obtain accurate SPR measurements, it is essential to analyze and mitigate potential noise sources that can influence the input signal and the transfer functions. This section identifies various noise sources, quantifies their impact on spectral precision, and describes the strategies employed to minimize their effects. This approach ensures precise SPR spectrum correction and reliable analyte analysis.

### 4.1. Noise-Source Characterization

*Light-Source Fluctuations:* Fluctuations in the intensity and spectral distribution in the lamp, despite using a regulated power supply, can introduce noise. The manufacturer specifies power stability below 0.05% and color temperature stability of ±15 K [20]. These fluctuations can affect the baseline stability of the SPR signal and the precision of resonance wavelength determination.*Dark Noise:* With the light source off, the spectrometer measured a normalized average intensity of $3\times 10^{-3}$. This low level, which is attributed to background noise and is unaffected by the activation of the ambient light, indicates that there are negligible contributions from dark current, electronic noise, and stray light. This verifies that the system design effectively mitigates these intrinsic sources of noise through its optical and electronic configuration.*Background Noise:* With the SLS201L light source activated, the background spectrum (Figure 8, dashed red line) exhibits a maximum normalized intensity of 1 in the central spectral region, declining to approximately 3 × 10^−3^ at the edges. This restricts the precise identification of plasmonic resonances at the spectral extremes, despite a maximum signal-to-noise ratio (SNR) of approximately 400.*Thermal Noise:* It is important to note that temperature variations have the potential to impact the output of the light source and the performance of the spectrometer detector. Both exhibit a temperature sensitivity of 0.1%/°C, as cited in reference [18,19,20]. The laboratory temperature is maintained at 20 degrees Celsius ±2 degrees Celsius. The combined uncertainty in light output and detector response resulting from this ±2 °C variation is approximately 0.28%, calculated by adding the individual uncertainties in quadrature.*Inherent Noise:* Molecular vibrations within the sample itself introduce inherent noise, causing fluctuations in the local refractive index and affecting the interaction with the evanescent wave. This noise impacts both the intensity and spectral resolution of the SPR signal and is considered in the analysis.

### 4.2. Mitigation Strategies and Their Effectiveness

*Light-Source Stabilization:* To mitigate light-source fluctuations, a regulated power supply was used, and the lamp was allowed to warm up for 10 min before measurements, following IES standards [33]. During this warm-up period, the spectrometer remained operational and exposed to the light source, allowing it to reach a stable thermal and electrical state. This procedure stabilized the luminous flux, ensuring a luminous intensity variation of less than 0.05% and a power variation limited to 0.01% per hour.*Background Noise Reduction:* The system design was optimized to minimize the intrusion of stray light. In this design, the elements are positioned as closely as possible. The collection optical fiber, with a numerical aperture of 0.22, was situated at a distance of 5 mm from the prism, which served to mitigate the influence of ambient light.*Thermal Stabilization:* The SPR-S system was housed in a temperature-controlled environment (18–22 °C) to minimize thermal noise and ensure stable and reliable measurements. This significantly reduced temperature-induced fluctuations in system performance.

### 4.3. Noise Quantification and Measurement Uncertainty

The performance of the SPR spectrometer system was evaluated by quantifying the signal-to-noise ratio (SNR) and spectral resolution. With an 8 ms integration time, the SNR exceeded 100 in the 500–900 nm range. Outside this range, the SNR decreased due to lower lamp irradiance and reduced CCD responsivity. The average resolution of the spectrometer was 0.22 nm, consistent with the manufacturer’s specifications [19]. The system sensitivity of $4\times 10^{-4}$ RIU/λ (refractive index unit per wavelength) [3,16] enabled a refractive index resolution of $8.8\times 10^{-5}$ RIU.

Figure 9 shows the calculated SNR for the measured spectrum, obtained using the ThorSpectra software, which controls and reads data from the spectrometer. A binomial smoothing filter was applied to both the signal and noise before calculating the SNR. The SNR varies with wavelength, peaking around 400. At the spectral extremes, below 500 nm and above 900 nm, the SNR falls below 100, as indicated by the dashed red line. This 100 SNR threshold was chosen based on experiments showing increased noise in these spectral regions, potentially impacting measurement accuracy. Applying the smoothing filter helps mitigate this noise and recover spectral information in the affected regions.

Considering all noise sources and mitigation strategies, the overall measurement uncertainty was estimated to be approximately 0.5%, calculated using the root sum square method. This uncertainty considers contributions from light-source stability, background noise, and thermal variations.

## 5. SPR Spectrum Correction

The accurate correction of the SPR spectrum is essential for reliable analyte analysis and precise molecular interaction studies. Current spectral correction methods often fall short of fully accounting for the wavelength-dependent performance of all system components, leading to discrepancies between experimental and theoretical spectra. This work introduces a novel correction method (Model 3) that addresses these limitations by comprehensively incorporating the transfer functions of the light source, polarizer, optical fibers, and spectrometer. This comprehensive approach results in a more accurate representation of the system’s spectral response, enabling the precise determination of the resonance wavelength and facilitating more reliable analysis of molecular interactions. The experimentally measured spectrum, $Y_{SPR}$, is often distorted by the wavelength-dependent performance of the system components. These distortions can shift the observed resonance wavelength and complicate spectral interpretation. To address this, a spectral correction function, C(λ), was developed, as shown in Figure 10a.

The correction function, C(λ), is derived by considering the combined effects of the system’s transfer functions. It is designed to fit the experimental spectra by accounting for the actual interactions of light with all system elements. The correction function is defined as:(12)$$C\left(\lambda \right)=\frac{1}{TF_{M3}+3Avg\left(Noise\right),}$$
where $TF_{M3}$ represents the system’s overall transfer function, as described in Section 3. Avg(Noise) represents the average noise in the system. The term 3Avg(Noise) is added to ensure robustness against unusual $TF_{M3}$ data, prevent abrupt changes in C(λ), avoid physically inconsistent negative values, and preclude division by zero. This value was empirically determined based on the stability of the corrected data.

The correction function, C(λ), represents the intrinsic spectral response of the measurement system, incorporating the experimentally determined transfer functions of the light source, polarizer, optical fibers, and spectrometer. This effectively removes the system’s spectral influence from the measured data.

The corrected spectrum, $Y_{cor}$, is then obtained by:(13)$$Y_{cor}=M\;Y_{SPR}\;C\left(\lambda \right),$$
where *M* is a scaling factor to adjust the overall amplitude of the spectrum, accounting for differences in integration time between sample and reference measurements. Applying this correction yields a spectrum that reflects the true resonance response of the SPR sensor, independent of the system’s spectral characteristics

For performance assessment, SPR-S measurements were performed using ultrapure water as the analyte due to its well-characterized refractive index, providing a reliable reference for system performance evaluation. The measured SPR spectrum ($Y_{SPR}$) and the simulated spectrum ($Y_{sim}$) were obtained, and the spectral correction process, using the previously defined C(λ), was applied. The results are illustrated in Figure 10b for an angle of incidence of 57.4°. The use of ultrapure water ensured no unexpected deposition on the gold substrate, guaranteeing results free from external interference.

The corrected spectrum ($Y_{cor}$) was compared to the theoretical spectrum ($Y_{sim}$) and spectra-corrected using two methods described by Liu et al. [7] for handling detector response variations:**Optimized Method:** Adds a compensation factor based on the normalized inverse of the Si detector response:(14)$$Y_{cor-Si}=Y_{SPR}+Y_{SPR}\cdot \left(1-H_{Spec}\left(\lambda \right)\right)$$**Routine Method:** Divides the experimental spectrum by the normalized detector response:(15)$$Y_{cor-routine}=\frac{Y_{SPR}}{H_{Spec}\left(\lambda \right)}$$

Figure 11 shows the results using these methods. While the optimized method shows some improvement over the routine method, neither matches the experimental curve as closely as our proposed model. The methods in [7] primarily focus on correcting spectral symmetry in the resonance region, whereas our model enhances the entire spectral profile. Therefore, $R^{2}$ and MSE were used as figures of merit for a comprehensive comparison.

Our model consistently outperforms both the optimized and routine methods from Liu et al., achieving an R² value of 98.3% and an MSE of 0.002. It is not necessary to calculate the figures of merit for the methods of Liu et al., as they are visually different and would lead to extreme values. Furthermore, the percentage error in resonance wavelength determination is substantially reduced with Model 3, reaching an average error of 0.4% compared to an average of 1.6% for the Liu et al. methods. This improvement highlights the advantage of Model 3’s comprehensive approach to spectral correction.

The results demonstrate that the correction function effectively compensates for spectral distortions, yielding high similarity between the corrected and theoretical spectra, not achieved by existing methods. This accurate spectral representation is crucial for reliable analysis of molecular interactions.

For unknown samples, the corrected spectrum, after calibration, can be used to determine properties like refractive index or concentration. It also facilitates monitoring molecular interactions, providing insights into nanoscale dynamic processes. Future work will explore applying this correction function to analyze nanosuspensions and dissolved molecules.

## 6. Conclusions

This work presents a novel approach for modeling and correcting the spectral response of a homemade surface-plasmon resonance spectrometer (SPR-S) using transfer functions (TFs). This method allows for a detailed understanding of the system’s behavior by characterizing the individual contributions of each optical component. Three models of increasing complexity were developed. Model 1 established a baseline, excluding optical fiber effects. Model 2 incorporated fiber TFs, highlighting their significant impact. Model 3, the most accurate, included a convolution operation to account for the combined effects of chromatic aberrations, spatial disturbances, and refractive index variations within the optical components. Model 3 achieved an R^2^ of 0.9842 and an MSE of 0.0018, effectively capturing the system’s spectral behavior.

Initially, the system’s operational range was limited to 400–700 nm due to the polarizer’s characterized range. However, by experimentally determining component TFs, the range was extended to 900 nm. This broadened spectral range is crucial for analyzing a wider variety of molecular interactions. Noise analysis and mitigation strategies resulted in an SNR exceeding 100 within the 500–900 nm range. The TFs also explained increased noise outside this range, attributed to lower lamp irradiance and CCD responsivity at the spectral extremes. Figure 9 shows the calculated SNR, with a demarcated threshold of 100.

A correction function, derived from Model 3’s TFs, was applied to correct measured SPR spectra. Validation with ultrapure water showed high agreement between corrected and theoretical spectra (similarity = 98.30%, errors < 1%). This correction is essential for accurate SPR analysis. Future work will apply this methodology to more complex analytes, like nanosuspensions and dissolved molecules, further exploring this approach’s potential for advanced SPR applications.

## Figures and Tables

**Figure 1 sensors-25-00894-f001:**
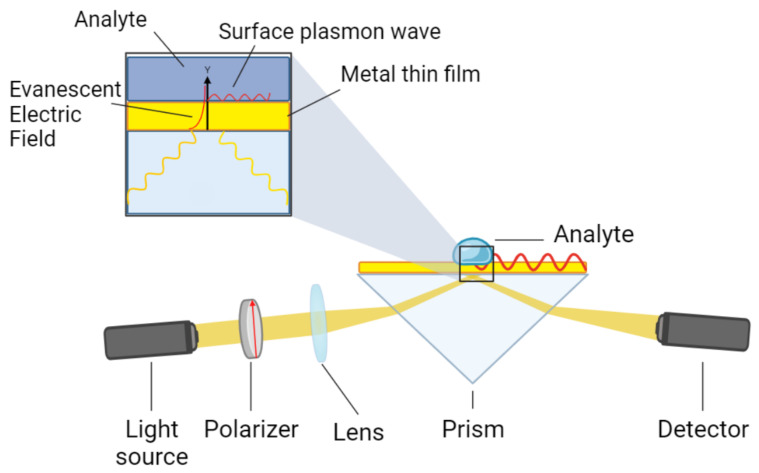
A scheme of the setup used for surface-plasmon resonance spectroscopy.

**Figure 2 sensors-25-00894-f002:**
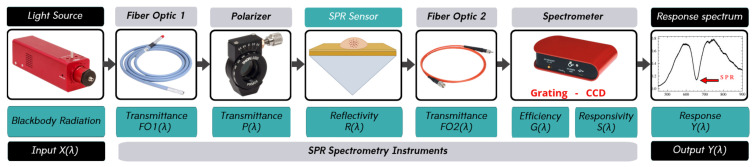
Measuring system instruments arranged as in the experiment. Y(λ) shows the result of the interaction of light with all components of the system.

**Figure 3 sensors-25-00894-f003:**
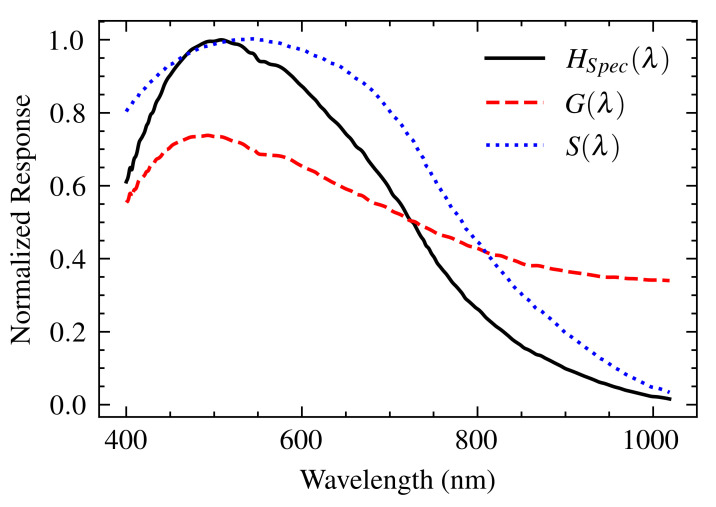
Transfer functions of the grating (G(λ), red dashed line) and the CCD sensor (S(λ), blue dotted line), resulting in the transfer function of the spectrometer.

**Figure 4 sensors-25-00894-f004:**
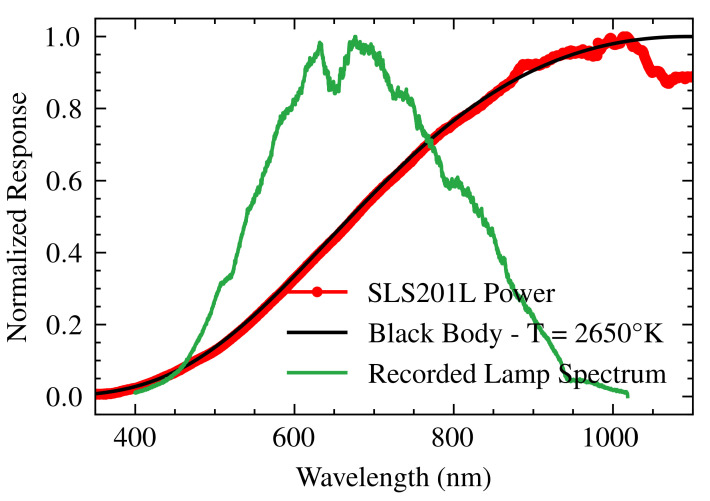
The spectrum of the lamp supplied by the manufacturer (red), fitted to the radiation of a blackbody (black), and recorded by the spectrometer (green).

**Figure 5 sensors-25-00894-f005:**
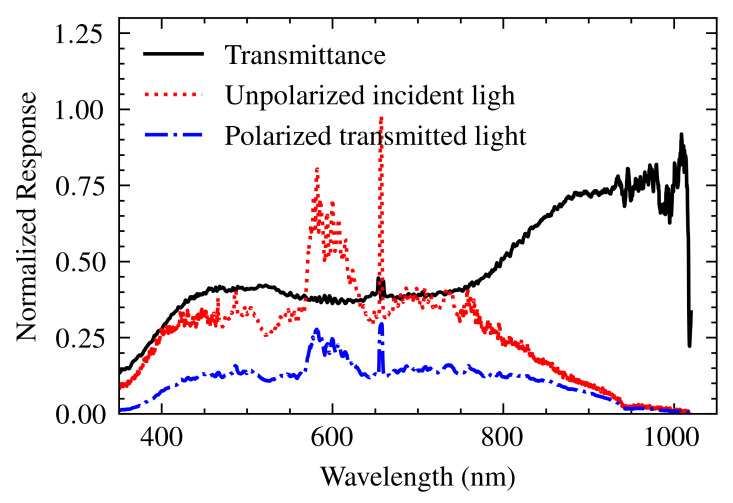
The experimentally determined transfer function (transmittance) of the polarizer, P(λ) (solid black line), derived from the incident unpolarized light spectrum (dotted red line) and the transmitted polarized light spectrum (dash-dotted blue line).

**Figure 6 sensors-25-00894-f006:**
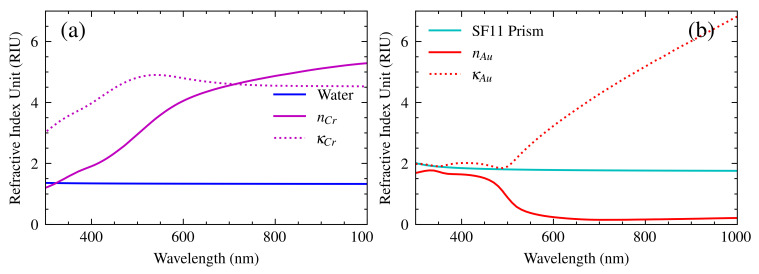
A refractive index of the materials that make up the SPR sensor divided into two graphs for easy identification. (**a**) Refractive index curves for water and chromium film. (**b**) Refractive index curves for SF11 and gold film. The glass and water indexes vary rather slowly in the spectral range depicted.

**Figure 7 sensors-25-00894-f007:**
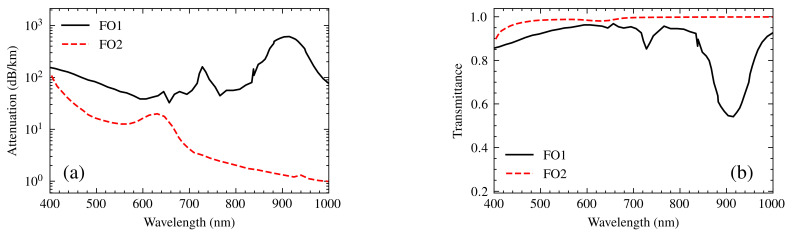
(**a**) Attenuation and (**b**) transmittance (transfer functions) of the optical fibers. The solid black line represents optical fiber 1 (illumination), while the dotted red line represents optical fiber 2 (collection).

**Figure 8 sensors-25-00894-f008:**
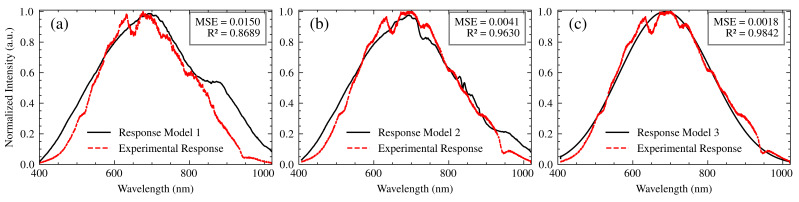
A comparison of the experimental and modeled spectra. The validation of the proposed models and an evaluation of their performance. (**a**) Model 1 excludes the influence of optical fibers. (**b**) Model 2 includes the transfer functions of optical fibers. (**c**) Model 3 includes a convolution operation to represent some optical phenomena.

**Figure 9 sensors-25-00894-f009:**
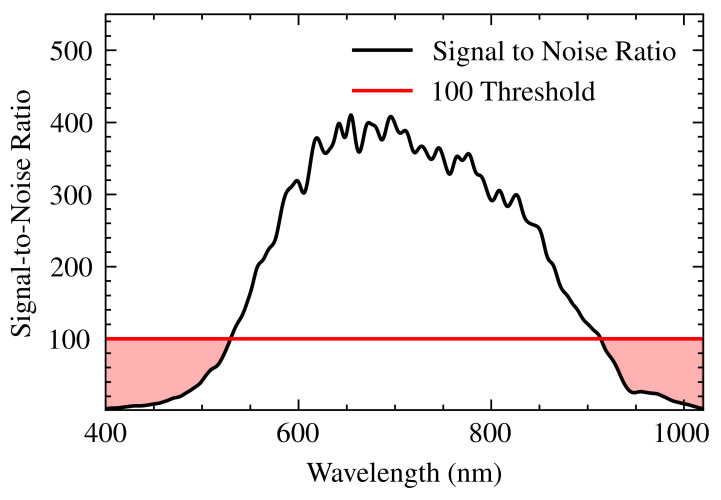
An SNR plot for the SPR-S system. The black curve (400–1000 nm) is smoothed with a binomial filter for clarity. The red line marks the 100 SNR threshold, highlighting the range where the system maintains high signal quality.

**Figure 10 sensors-25-00894-f010:**
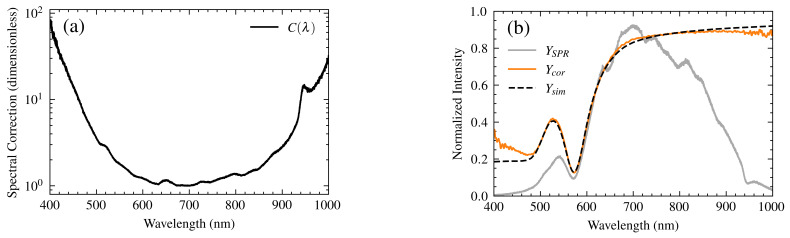
Spectrum correction and comparison function. (**a**) Correction function, C(λ), on a logarithmic scale. (**b**) A comparison of the experimental, corrected, and theoretical spectra for ultrapure water at an incidence angle of 57.4°. Experimental spectrum ($Y_{SPR}$, gray line); corrected spectrum ($Y_{cor}$, orange line); and theoretical spectrum ($Y_{sim}$, black dashed line).

**Figure 11 sensors-25-00894-f011:**
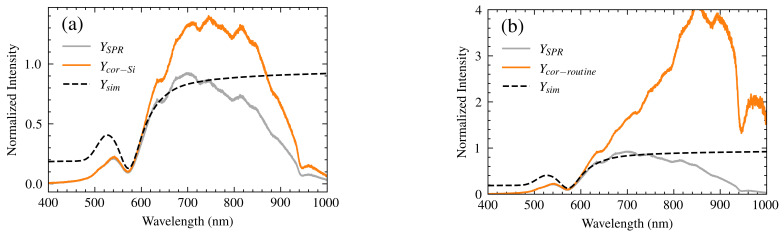
The correction of spectra using the models described by Tingting et al. (**a**) Routine model and (**b**) proposed model for silicon sensors. In both graphs, the experimental spectrum ($Y_{SPR}$, gray line); corrected spectrum ($Y_{cor}$, orange line); and theoretical spectrum ($Y_{sim}$, black dashed line) are shown.

## Data Availability

All data are included in the article.

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
