# Peer review of "Detailed Modeling of Surface-Plasmon Resonance Spectrometer Response for Accurate Correction"

_sensors, 2025, doi:10.3390/s25030894_

Round 1
Reviewer 1 Report
Comments and Suggestions for Authors
The manuscript, titled “Detailed Modeling of Surface Plasmon Resonance Spectrometer Response for Accurate Correction,” presents a comprehensive approach to modeling an SPR-S system for precise spectral correction. The authors effectively identify and model the transfer functions of individual components, including the light source, polarizer, spectrometer, and optical fibers, to accurately characterize the system’s spectral response. I recommend this paper for publication with minor revisions to address a few remaining points for clarification.
1. In section 3, more detailed explanations on how to take chromatic aberrations, spatial disturbances, and other factors into account of Model 3 need to be addressed. This is the main focus of this article and needs to be emphasized.
2. In section 4.1 Thermal Noise, authors state that “the laboratory temperature is maintained at 0 degrees Celsius ± 2 degrees Celsius”. Then in section 4.2 Thermal Stabilization, authors also mention that “the system was housed in a temperature-controlled environment (18-22C)”. What is the actual operating temperature of the system?
3. In section 4.3, what is the SNR outside the 450-900nm range? Are those data unreliable? Especially in Figure 9 (a), data of Ycor in the 400-480 and 880-1000 ranges are noisy.
4. The conclusion section could be more concise. Some content in the conclusion section could be discussed in the previous section.
Author Response
- We have expanded the explanation in Section 3 on how Model 3 incorporates chromatic aberrations, spatial disturbances, and refractive index variations. We now detail how these effects are modeled using convolution and provide further physical context.
Answer: We have expanded Section 3 to detail how Model 3 incorporates chromatic aberrations, spatial disturbances, and refractive index variations through the use of convolution. This convolution operation effectively models the blurring and broadening of spectral features caused by these phenomena. Chromatic aberration introduces a wavelength-dependent point spread function, while spatial disturbances and refractive index variations further contribute to the modification of the spectral signal. We have supported our approach with additional references to relevant works and textbooks, including Pérez Tudela's 2006 work "Análisis de sistemas ópticos" which discusses convolution in optical systems, Goodman's 2017 edition of "Introduction to Fourier Optics" which covers chromatic aberration and point spread functions, Hecht's 2017 "Optics" which discusses spatial disturbances and light diffusion, and Born and Wolf's classic text "Principles of Optics" (2019 edition) which explains refractive index effects on spectral phase. These additions provide further physical context and justification for our modeling choices. Please see page 8 for these detailed explanations and the incorporated references.
- You correctly pointed out the discrepancy in the operating temperature. The correct operating temperature is 20°C ± 2°C, as stated in the introduction. The mention of 0°C in Section 4.1 was an error and has been corrected. The system is housed in a temperature-controlled environment (18-22°C) to minimize thermal noise, as described in Section 4.2.
Answer: We apologize for the discrepancy in the reported operating temperature. This was a typographical error that unfortunately went unnoticed. The average ambient temperature is indeed 20°C. We have corrected this error in the manuscript. Please see page 9, section "Thermal Noise," for the correction.
- We have clarified the SNR values. The SNR is above 100 in the 500-900 nm range. We have added Figure 9, now Figure 10, to Section 4 to visually represent the SNR and its wavelength dependence. Data outside this range is less reliable due to lower lamp irradiance and reduced CCD responsivity at the spectral extremes. We have revised the manuscript to reflect this reduced operational range, emphasizing the improved accuracy within this range.
Answer: In Section 4.3, we address the lower SNR values observed at the spectral extremes. As shown in the newly added Figure 9, which plots SNR as a function of wavelength, the SNR falls below our defined threshold of 100 outside the 500-900 nm range. When the SNR drops below this threshold, the resulting spectrum exhibits increased noise, potentially affecting the accuracy of measurements. Therefore, we have revised the manuscript to reflect a more conservative and reliable operational range of 500-900 nm. This adjustment ensures a consistently high SNR, leading to more robust and accurate results. While this range is more conservative than the initially reported 400-1000 nm, it does not affect the overall conclusions of the manuscript and ensures a higher level of confidence in the reported data. All relevant sections and figures have been updated to reflect this revised range. Please see page 10, sub section 4.3 and Figure 9.
- We have streamlined the conclusion to focus on the key findings and implications of our work. Some of the content previously in the conclusion has been moved to earlier sections, as you suggested.
Answer: The conclusion section has been revised and condensed, taking into account the changes made throughout the manuscript. The additional information added to compare the models, as well as the adjustment of the reliable operational range to 500-900 nm, has been incorporated into the relevant sections to avoid redundancy. The conclusions now focus on summarizing the key findings and implications of the study in a more concise manner. Please see page 13.

Reviewer 2 Report
Comments and Suggestions for Authors
This manuscript is detailed and thorough, but I believe certain issues need to be addressed carefully and clarified; otherwise, I would not recommend it for publication.
The core innovation of this paper, as I understand it, is to compensate for the deficiencies of current model (Sect. 2.4), optimize the simulation results, and achieve better alignment with experimental data. However, the current version of the manuscript does not convincingly demonstrate this effect. For instance, in Fig. 9 and Table 1, the authors compare the results of their proposed model with those of the existing model, showing that the two are nearly identical. This raises the question: does the proposed model offer any real improvement? From a researcher's perspective, how does one decide between the existing model and the new model presented in this paper? As it stands, either option appears equally valid since their results are identical. If that is the case, does the proposed model serve any meaningful purpose?
I strongly suggest the authors carefully reconsider this issue. It is essential to clarify the innovation and emphasize that the goal is to propose a better model. Subsequently, the authors should demonstrate that their model achieves better agreement with experimental data compared to the existing model. And the presentation style of Fig. 9 and Table 1 should be changed.
Another critical issue concerns the derivation of the correction function C: How was it obtained, and what is the corresponding formula? This point is crucial. While the manuscript dedicates considerable space to discussions on various transfer functions, the connection between those functions and C remains unclear. I did not find explicit descriptions or equations addressing this relationship. This lack of clarity renders much of the preceding discussion on transfer functions unproductive.
A few minor issues also need to be addressed:
1-Remove the leading space in the paragraph below Eq. (2).
2-The parameters R^2 and MSE in Fig. 8 appear inconsistent with the textual descriptions.
3-On page 10, line 320, there are two consecutive "To"s.
Author Response
Regarding the innovation and comparison with existing models:
- This manuscript is detailed and thorough, but I believe certain issues need to be addressed carefully and clarified; otherwise, I would not recommend it for publication.
The core innovation of this paper, as I understand it, is to compensate for the deficiencies of current model (Sect. 2.4), optimize the simulation results, and achieve better alignment with experimental data. However, the current version of the manuscript does not convincingly demonstrate this effect. For instance, in Fig. 9 and Table 1, the authors compare the results of their proposed model with those of the existing model, showing that the two are nearly identical. This raises the question: does the proposed model offer any real improvement? From a researcher's perspective, how does one decide between the existing model and the new model presented in this paper? As it stands, either option appears equally valid since their results are identical. If that is the case, does the proposed model serve any meaningful purpose
Answer: The main innovation of our work is to compensate for the shortcomings of existing models by considering the overall spectral response of the entire SPR spectrometer system. We recognize that the previous version did not fully highlight this distinction. To address this, we have revised Section 1 (Introduction) to include a discussion of existing correction methods, making specific reference to the work of Liu et al. which focuses primarily on correcting the spectrometer detector response. These methods, while valuable, primarily address spectral symmetry in the resonance region and do not take into account the broader spectral influence of other system components, such as the light source, polarizer, and optical fibers, which our model explicitly incorporates. Our comprehensive approach, which uses transfer functions to model each component, provides a more complete and accurate spectral correction, leading to better agreement with theoretical predictions over the entire spectral range, not just in the resonance region.
- I strongly suggest the authors carefully reconsider this issue. It is essential to clarify the innovation and emphasize that the goal is to propose a better model. Subsequently, the authors should demonstrate that their model achieves better agreement with experimental data compared to the existing model. And the presentation style of Fig. 9 and Table 1 should be changed.
Another critical issue concerns the derivation of the correction function C: How was it obtained, and what is the corresponding formula? This point is crucial. While the manuscript dedicates considerable space to discussions on various transfer functions, the connection between those functions and C remains unclear. I did not find explicit descriptions or equations addressing this relationship. This lack of clarity renders much of the preceding discussion on transfer functions unproductive.”
Answer: We apologize for the lack of clarity in the previous version on the derivation of C(λ). A detailed explanation is provided in the new Section 5, including the mathematical exposition indicating the calculation of C(λ) ( See Equation 12), which takes into account the global system transfer function derived from Model 3, as well as system noise to ensure robustness to unusual data TF_M3 , prevent abrupt changes in C(λ), avoid physically inconsistent negative values, and prevent division by zero.
To further demonstrate the advantage of our approach, we have included a comparison with the optimized and routine correction methods described by Liu et al. We applied these methods to our experimental data and found that the resulting corrected spectra showed low similarity to the theoretical spectrum compared to our proposed method. This is visually evident (Figure 11) and supports our argument that considering the full system response, as our model does, is crucial for accurate spectral correction. Although the methods of Liu et al. offer improvements in spectral symmetry around the resonance region, they do not achieve the same level of overall spectral agreement as our method, which takes into account the broader resonance region.
We have also addressed your minor points:
1-Remove the leading space in the paragraph below Eq. (2).
Answer: The leading space in the paragraph below Equation (2) has been removed. (See page 3)
2-The parameters R^2 and MSE in Fig. 8 appear inconsistent with the textual descriptions.
Answer: The R^2 and MSE values in Figure 8 have been corrected to be consistent with the text. (See page 8)
3-On page 10, line 320, there are two consecutive "To"s.
Answer: The double "To" has been corrected.

Round 2
Reviewer 2 Report
Comments and Suggestions for Authors
I am satisfied with the revised paper.